# VoxDet: Voxel Learning for Novel Instance Detection

**Bowen Li**[1]     **Jiashun Wang**[1]     **Yaoyu Hu**[1]     **Chen Wang**[2]     **Sebastian Scherer**[1]

[1] Carnegie Mellon University     [2] State University of New York at Buffalo

## Abstract

Detecting unseen instances based on multi-view templates is a challenging problem due to its open-world nature. Traditional methodologies, which primarily rely on 2D representations and matching techniques, are often inadequate in handling pose variations and occlusions. To solve this, we introduce VoxDet, a pioneer 3D geometry-aware framework that fully utilizes the strong 3D voxel representation and reliable voxel matching mechanism. VoxDet first ingeniously proposes template voxel aggregation (TVA) module, effectively transforming multi-view 2D images into 3D voxel features. By leveraging associated camera poses, these features are aggregated into a compact 3D template voxel. In novel instance detection, this voxel representation demonstrates heightened resilience to occlusion and pose variations. We also discover that a 3D reconstruction objective helps to pre-train the 2D-3D mapping in TVA. Second, to quickly align with the template voxel, VoxDet incorporates a Query Voxel Matching (QVM) module. The 2D queries are first converted into their voxel representation with the learned 2D-3D mapping. We find that since the 3D voxel representations encode the geometry, we can first estimate the relative rotation and then compare the aligned voxels, leading to improved accuracy and efficiency. In addition to method, we also introduce the first instance detection benchmark, RoboTools, where 20 unique instances are video-recorded with camera extrinsic. RoboTools also provides 24 challenging cluttered scenarios with more than 9k box annotations. Exhaustive experiments are conducted on the demanding LineMod-Occlusion, YCB-video, and RoboTools benchmarks, where VoxDet outperforms various 2D baselines remarkably with faster speed. To the best of our knowledge, VoxDet is the first to incorporate implicit 3D knowledge for 2D novel instance detection tasks. Our code, data, raw results, and pre-trained models are public at `https://github.com/Jaraxxus-Me/VoxDet`.

## 1 Introduction

Consider the common scenarios of locating the second sock of a pair in a pile of laundry or identifying luggage amid hundreds of similar suitcases at an airport. These activities illustrate the remarkable capability of human cognition to swiftly and accurately identify a specific *instance* among other similar objects. Humans can rapidly create a mental picture of a novel *instance* with a few glances even if they see such an *instance* for the first time or have never seen *instances* of the same type. Searching for instances using mental pictures is a fundamental ability for humans, however, even the latest object detectors [1–7] still cannot achieve this task.

We formulate the above tasks as novel instance detection, that is identification of an unseen instance in a cluttered query image, utilizing its multi-view support references. Previous attempts mainly work in 2D space, such as correlation [8, 5], attention mechanisms [6], or similarity matching [9], thereby localizing and categorizing the desired instance, as depicted in Fig. 1 gray part. However, these techniques struggle to maintain their robustness when faced with significant disparities between the query and templates. In comparison to novel instance detection, there is a vast amount of work centered around few-shot category-level object detection [7, 1, 2]. Yet, these class-level matching techniques prove insufficient when it comes to discerning specific instance-level features.

37th Conference on Neural Information Processing Systems (NeurIPS 2023).

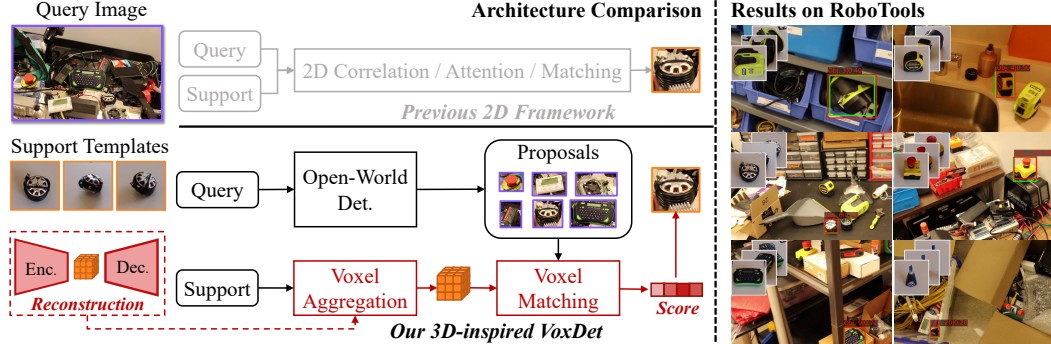

Figure 1: Architecture comparison between previous 2D methods (*gray*) and proposed VoxDet (**black**). Previous methods resorts to pure 2D correlation/attention/matching for novel instance detection. In contrast, VoxDet is 3D-inspired, leveraging reconstruction objective to learn the geometry-aware voxel representation, which enables more effective and accurate voxel-based instance detection. In the challenging newly built RoboTools benchmark shown on the right, VoxDet exhibits surprising robustness to severe occlusion and orientation variation.

Humans exhibit the remarkable capability to swiftly formulate a mental model of an unfamiliar instance, facilitated by a rapid comprehension of its 3D geometric structure [10–12]. Leveraging such a mental representation, once presented with a single query image, a human can probably search and identify the same instance despite alterations in distance, occlusion, and even approximate the instance's orientation. Motivated by this, we propose VoxDet, a pioneer 3D geometry-aware instance detection framework as shown in Fig. 1 bottom. In contrast to state-of-the-art methods [7, 5, 6, 9, 13], VoxDet adapts two novel designs: (1) a compact 3D voxel representation that is robust to occlusion and pose variations and (2) an effective voxel matching algorithm for identifying instances.

VoxDet consists of three main modules: a template voxel aggregation (TVA) module, an open-world detection module, and a query voxel matching (QVM) module. Initially, the TVA module transforms multi-view 2D features of an instance into individual 3D template voxels [10]. These template voxels are then accumulated using relative rotations, thus incorporating both geometry and appearance into a condensed template voxel. As VoxDet learns this 2D-3D mapping via a reconstruction objective, TVA effectively encapsulates both the geometry and appearance of any instance into a compact template voxel. When presented with a query image, VoxDet employs an open-world detector [14] that universally identifies potential objects within the image as 2D proposals. These proposals are then converted to query voxels via the learned 2D-3D mapping and compared with the template voxel by the QVM module. QVM initiates this comparison process by first estimating the relative rotation between a query voxel and the template, which is then used to align the two voxels. Finally, the comparison between aligned voxels is delivered by a carefully designed voxel relation module.

Besides methodology, we also construct a large-scale synthetic training dataset, Open-World Instance Detection (OWID). OWID comprises 10k instances sourced from the ShapeNet [15] and Amazon Berkeley Objects [16] datasets, culminating in 55k scenes and 180k query bounding boxes. Trained on OWID, VoxDet demonstrates strong generalization ability on novel instances, which we attribute to the meticulously designed voxel-based framework and the large-scale OWID training set.

To validate VoxDet, we further build RoboTools, a new instance detection benchmark compiled from a diverse range of real-world cluttered environments. RoboTools consists of 20 unique instances, 24 test scenes, and over 9,000 annotated bounding boxes. As shown in Fig. 1 right, in the demanding RoboTools benchmark, VoxDet can robustly detect the novel instances under severe occlusion or varied orientation. Evaluations are also performed on the authoritative Linemod-Occlusion [17] and YCB-video [18] for more compelling results. The exhaustive experiments on these three benchmarks demonstrate that our 3D geometry-aware VoxDet not only outperforms various previous works [5–7] and different 2D baselines [19, 9] but also achieves faster inference speed.

## 2 Related Works

**Typical object detection** [20–26] thrive in category-level tasks, where all the instances belonging to a pre-defined class are detected. Typical object detection can be divided into two-stage approaches and

one-stage approaches. For the former one, RCNN [20] and its variants [21, 22] serves as foundations, where the regions of interest (ROI) are first obtained by the region proposal network. Then the detection heads classify the labels of each ROI and regress the box coordinates. On the other hand, the YOLO series [23–25] and recent transformer-based methods [4, 3] are developing promisingly as the latter stream, where the detection task is tackled as an end-to-end regression problem.

**Few-shot/One-shot object detection** [1, 27, 28, 2, 29, 7] can work for unseen classes with only a few labeled support samples, which are closer to our task. One stream focuses on transfer-learning techniques [28, 27], where the fine-tuning stage is carefully designed to make the model quickly generalize to unseen classes. While the other resorts to meta-learning strategies [1, 7, 2, 29], where various kinds of relations between supports and queries are discovered and leveraged. Since the above methods are category-level, they assume more than one desired instances exist in an image, so the classification/matching designs are usually tailored for Top-100 precision, which is not a very strict metric. However, they can easily fail in our problem, where the *Top-1* accuracy is more important.

**Open-world/Zero-shot object detection** [30–32, 14] finds any objects on an image, which is class-agnostic and universal. Some of them learn objectiveness [30, 14] and others [32] rely on large-scale high-quality training sets. These methods can serve as the first module in our pipeline, which generates object proposals for comparison with the templates. Among them, we adopt [14] with its simple structure and promising performance.

**Instance detection** requires the algorithm to find an unseen instance in the test image with some corresponding templates. Previous methods [6, 5, 8] usually utilize pure 2D representations and 2D matching/relation techniques. For example, DTOID [6] proposed global object attention and a local pose-specific branch to predict the template-guided heatmap for detection. However, they easily fall short when the 2D appearance variates due to occlusion or pose variation. Differently, VoxDet leverages the explicit 3D knowledge in the multi-view templates to represent and match instances, which is geometry-invariant.

**Multi-view 3D representations** Representing 3D scenes/instances from multi-view images is a long-standing problem in computer vision. Traditional methods resort to multi-view geometry, where structure from motion (SfM) [33] pipeline has enabled joint optimization of the camera pose and 3D structure. Modern methods usually adopts neural 3D representations [34, 11, 35–37, 12, 10], including deep voxels [35, 12, 10, 38] and implicit functions [36, 37], which have yielded great success in 3D reconstruction or novel view synthesis. Our framework is mainly inspired by Video Autoencoder [10], which encodes a video by separately learning the deep implicit 3D structure and the camera trajectory. One biggest advantage of [10] is that the learned Autoencoder can encode and synthesize test scenes without further tuning or optimization, which greatly satisfies the efficiency requirement of our instance detection task.

## 3 Methodology

### 3.1 Problem Formulation

Given a training instance set $\mathcal{O}_{\text{base}}$ and an unseen test instance set $\mathcal{O}_{\text{novel}}$, where $\mathcal{O}_{\text{base}} \cap \mathcal{O}_{\text{novel}} = \phi$, the task of novel instance detection (open-world detection) is to find an instance detector trained on $\mathcal{O}_{\text{base}}$ and then detect new instances in $\mathcal{O}_{\text{novel}}$ with no further training or finetuning. Specifically, for each instance, the input to the detector is a query image $\mathcal{I}^Q \in \mathbb{R}^{3 \times W \times H}$ and a group of $M$ support templates $\mathcal{I}^S \in \mathbb{R}^{M \times 3 \times W \times H}$ of the target instance. The detector is expected to output the bounding box $\mathbf{b} \in \mathbb{R}^4$ of an instance on the query image. We assume there exists exactly one such instance in the query image and the instance is located near the center of the support images.

### 3.2 Architecture

The architecture of VoxDet is shown in Fig. 2, which consists of an open-world detector, a template voxel aggregation (TVA) module, and a query voxel matching (QVM) module. Given the query image, the open-world detector aims to generate universal proposals covering all possible objects. TVA aggregates multi-view supports into a compact template voxel via the relative camera pose between frames. QVM lifts 2D proposal features onto 3D voxel space, which is then aligned and matched with the template voxel. In order to empower the voxel representation with 3D geometry, we first resort to a reconstruction objective in the first stage. The pre-trained models serve as the initial weights for the second instance detection training stage.

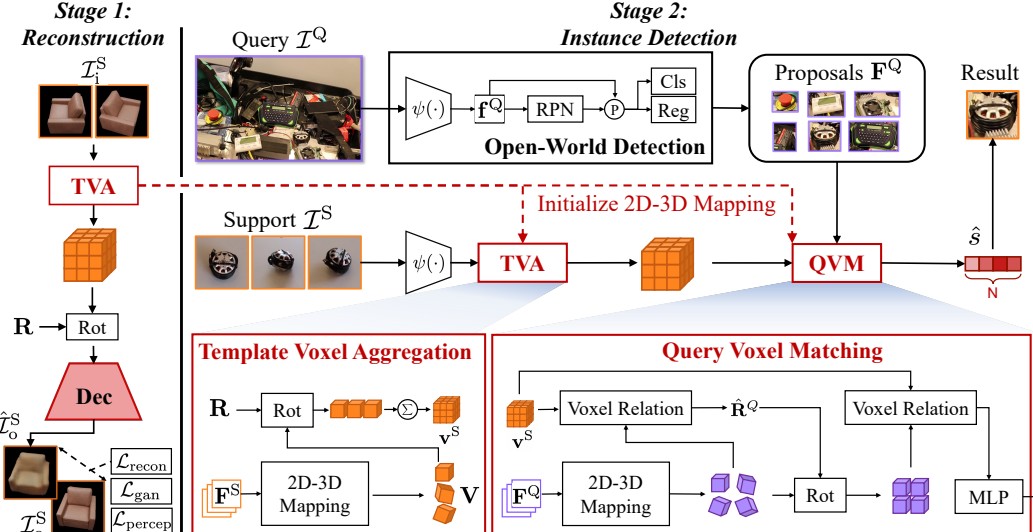

Figure 2: Architecture of VoxDet. VoxDet mainly consists of three modules, namely, open-world detection, template voxel aggregation (TVA), and query voxel matching (QVM). We first train TVA via the reconstruction stage, where the 2D-3D mapping learns to encode instance geometry. Then the pre-trained mapping serves as initial weights in the TVA and QVM modules for detection training.

### 3.2.1   Open-World Detection

Since the desired instance is unseen during training, directly regressing its location and scale is non-trivial. To solve this, we first use an open-world detector [14] to generate the most possible candidates. Different from standard detection that only finds out pre-defined classes, an open-world detector locates *all* possible objects in an image, which is class-agnostic.

As shown in Fig. 2, given a query image $\mathcal{I}^Q$, a 2D feature map $\mathbf{f}^Q$ is extracted by a backbone network $\psi(\cdot)$. To classify each pre-defined anchor as foreground (objects) or background, the region proposal network (RPN) [22] is adopted. Concurrently, the boundaries of each anchor are also roughly regressed. The resulting anchors with high classification scores are termed region proposals $\mathbf{P} = [\mathbf{p}_1, \mathbf{p}_2, \cdots, \mathbf{p}_N] \in \mathbb{R}^{N \times 4}$, where $N$ is the number of proposals. Next, to obtain the features $\mathbf{F}^Q$ for these candidates, we use region of interest pooling (ROIAlign) [22], $\mathbf{F}^Q = \text{ROIAlign}(\mathbf{P}, \mathbf{f}^Q) \in \mathbb{R}^{N \times C \times w \times w}$, where $C$ denotes channel dimensions and $w$ is the spatial size of proposal features. Finally, we obtain the final classification result and bounding box by two parallel multi-layer perceptrons (MLP), known as the detection head, which takes the proposal features $\mathbf{F}^Q$ as input, and outputs the binary classification scores and the box regression targets. The training loss is comprised of RPN classification loss $\mathcal{L}_{\text{cls}}^{\text{RPN}}$, RPN regression loss $\mathcal{L}_{\text{reg}}^{\text{RPN}}$, head classification loss $\mathcal{L}_{\text{cls}}^{\text{Head}}$, and head regression loss $\mathcal{L}_{\text{reg}}^{\text{Head}}$.

To make the detector work for open-world objects, the classification branches (in RPN and head) are guided by *objectiveness regression* [14]. Specifically, the classification score is defined (supervised) by Intersection over Union (IoU), which showed a high recall rate over the objects in test images, even those unseen during training. Since they have learned the class-agnostic "objectiveness", we assume the open-world proposals probably cover the desired novel instance. Therefore, we take the top-ranking candidates and their features as the input of the subsequent matching module.

### 3.2.2   Template Voxel Aggregation

To learn geometry-invariant representations, the Template Voxel Aggregation (TVA) module compresses multi-view 2D templates into a compact deep voxel. Inspired by previous technique [10] developed for unsupervised video encoding, we propose to encode our instance templates via their relative orientation in the physical 3D world. To this end, we first generate the 2D feature maps $\mathbf{F}^S = \psi(\mathcal{I}^S) \in \mathbb{R}^{M \times C \times w \times w}$ using a shared backbone network $\psi(\cdot)$ used in the query branch and then map the 2D features to 3D voxels for multi-view aggregation.

**2D-3D mapping:** To map these 2D features onto a shared 3D space for subsequent orientation-based aggregation, we utilize an implicit mapping function $\mathcal{M}(\cdot)$. This function translates the 2D features to 3D voxel features, denoted by $\mathbf{V} = \mathcal{M}(\mathbf{F}^S) \in \mathbb{R}^{M \times C_v \times D \times L \times L}$, where $\mathbf{V}$ is the 3D voxel feature

from the 2D feature, $C_v$ is the feature dimension, and $D, L$ indicate voxel spatial size. Specifically, we first reshape the feature maps to $\mathbf{F}'^{S} \in \mathbb{R}^{M \times (C/d) \times d \times w \times w}$, where $d$ is the pre-defined implicit depth, then we apply 3D inverse convolution to obtain the feature voxel.

Note that with multi-view images, we can calculate the relative camera rotation easily via Structure from Motion (SfM) [33] or visual odometry [39]. Given that the images are object-centered and the object stays static in the scene, these relative rotations in fact represent the relative rotations between the object orientations defined in the same camera coordination system. Different from previous work [10] that implicitly learns the camera extrinsic for unsupervised encoding, we aim to explicitly embed such geometric information. Specifically, our goal is to first transform every template into the same coordinate system using their relative rotation, which is then aggregated:

$$\mathbf{v}^{S} = \frac{1}{M} \sum_{i=1}^{M} \text{Conv3D}(\text{Rot}(\mathbf{V}_i, \mathbf{R}_i^{\top})) , \tag{1}$$

where $\mathbf{V}_i \in \mathbb{R}^{C_v \times D \times L \times L}$ is the previously mapped $i$-th independent voxel feature, $\mathbf{R}_i^{\top}$ denotes the relative camera rotation between the $i$-th support frame and the first frame. $\text{Rot}(\cdot, \cdot)$ is the 3D transform used in [10], which first wraps a unit voxel to the new coordination system using $\mathbf{R}_i^{\top}$ and then samples from the feature voxel $\mathbf{V}_i$ with the transformed unit voxel grid. Therefore, all the $M$ voxels are transformed into the same coordinate system defined in the first camera frame. These are then aggregated through average pooling to produce the compact template voxel $\mathbf{v}^{S}$.

By explicitly embedding the 3D rotations into individual reference features, TVA achieves a geometry-aware compact representation, which is more robust to occlusion and pose variation.

### 3.2.3 Query Voxel Matching

Given the proposal features $\mathbf{F}^{Q}$ from query image $\mathcal{I}^{Q}$ and the template voxel $\mathbf{C}^{S}$ from supports $\mathcal{I}^{S}$, the task of the query voxel matching (QVM) module is to classify each proposal as foreground (the reference instance) or background. As shown in Fig. 2, in order to empower the 2D features with 3D geometry, we first use the same mapping to get query voxels, $\mathbf{V}^{Q} = \mathcal{M}(\mathbf{F}^{Q}) \in \mathbb{R}^{N \times C_v \times D \times L \times L}$. VoxDet next accomplishes matching $\mathbf{v}^{S}$ and $\mathbf{V}^{Q}$ through two steps. First, we need to estimate the relative rotation between query and support, so that $\mathbf{V}^{Q}$ can be aligned in the same coordinate system as $\mathbf{v}^{S}$. Second, we need to learn a function that measures the distance between the aligned two voxels. To achieve this, we define a voxel relation operator $\mathcal{R}_v(\cdot, \cdot)$:

**Voxel Relation** Given two voxels $\mathbf{v}_1, \mathbf{v}_2 \in \mathbb{R}^{c \times a \times a \times a}$, where $c$ is the channel and $a$ is the spatial dimension, this function seeks to discover their relations in every semantic channel. To achieve this, we first interleave the voxels along channels as $\text{In}(\mathbf{v}_1, \mathbf{v}_2) = [\mathbf{v}_1^1, \mathbf{v}_2^1, \mathbf{v}_1^2, \mathbf{v}_2^2, \cdots, \mathbf{v}_1^c, \mathbf{v}_2^c] \in \mathbb{R}^{2c \times a \times a \times a}$, where $\mathbf{v}_1^k, \mathbf{v}_2^k$ is the voxel feature in the $k$-th channel. Then, we apply grouped convolution as $\mathcal{R}_v(\mathbf{v}_1, \mathbf{v}_2) = \text{Conv3D}(\text{In}(\mathbf{v}_1, \mathbf{v}_2), \text{group} = c)$. In the experiments, we found that such a design makes relation learning easier since each convolution kernel is forced to learn the two feature voxels from the same channel. With this voxel relation, we can then roughly estimate the rotation matrix $\hat{\mathbf{R}}^{Q} \in \mathbb{R}^{N \times 3 \times 3}$ of each query voxel relative to the template as:

$$\hat{\mathbf{R}}^{Q} = \text{MLP}(\mathcal{R}_v(\mathbf{V}^{S}, \mathbf{V}^{Q})) , \tag{2}$$

where $\mathbf{v}^{S}$ is copied $N$ times to get $\mathbf{V}^{S}$. In practice, we first predict 6D continuous vector [40] as the network outputs and then convert the vector to a rotation matrix. Next, we can define the classification haed with the Voxel Relation as:

$$\hat{s} = \text{MLP}\left(\mathcal{R}_v(\mathbf{V}^{S}, \text{Rot}(\mathbf{V}^{Q}, \hat{\mathbf{R}}^{Q}))\right) , \tag{3}$$

where $\text{Rot}(\mathbf{V}^{Q}, \hat{\mathbf{R}}^{Q})$ rotates the queries to the support coordination system to allow for reasonable matching. In practice, we additionally introduced a global relation branch for the final score, so that the lost semantic information in implicit mapping can be retrieved. More details are available in the supplementary material. During inference, we rank the proposals $\mathbf{P}$ according to their matching score and take the Top-k candidates as the predicted box $\hat{\mathbf{b}}$.

### 3.3 Training Objectives

As illustrated in Fig. 2, VoxDet contains two training stages: reconstruction and instance detection.
**Reconstruction** To learn the 3D geometry relationships, specifically 3D rotation between instance templates, we pre-train the implicit mapping function $\mathcal{M}(\cdot)$ using a reconstruction objective. We

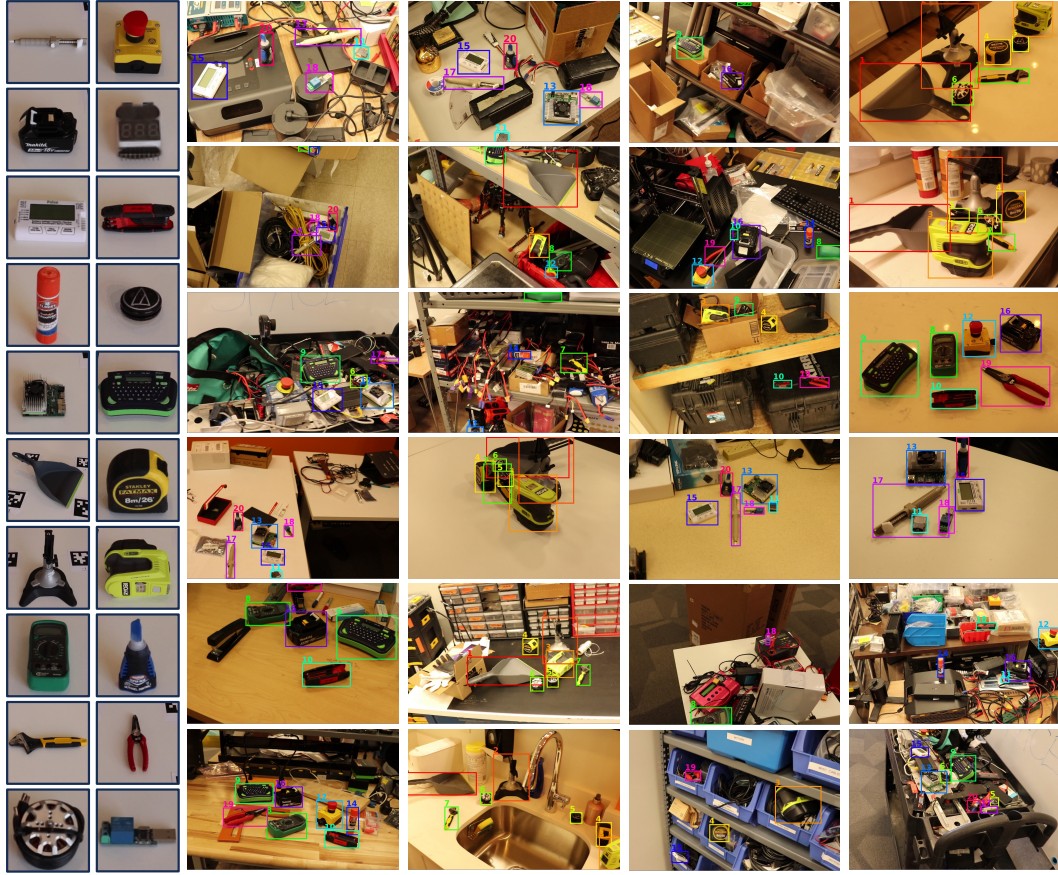

Figure 3: The instances and test scenes in the newly built RoboTools benchmark. The 20 unique instances are recorded as multi-view videos, where the relative camera poses between frames are provided. RoboTools consists of various challenging scenarios, where the desired instance could be under severe occlusion or in different orientation.

divide $M$ multi-view templates $\mathcal{I}^S$ into input images $\mathcal{I}^S_i \in \mathbb{R}^{(M-K)\times 3\times W\times H}$ and outputs $\mathcal{I}^S_o \in \mathbb{R}^{K\times 3\times W\times H}$. Next, we construct the voxel representation $\mathbf{V}^S$ using $\mathcal{I}^S_i$ via the TVA module and adopt a decoder network Dec to reconstruct the output images through the relative rotations:

$$\hat{\mathcal{I}}^S_{o,j} = \mathrm{Dec}(\mathrm{Rot}(\mathbf{V}^S, \mathbf{R}^\top_j))\,, j\in\{1,2,\cdots,K\}\,, \tag{4}$$

where $\hat{\mathcal{I}}^S_{o,j}$ denotes the $j$-th reconstructed (fake) output images and $\mathbf{R}_j$ is the relative rotation matrix between the 1-st to $j$-th camera frame. We finally define the reconstruction loss as:

$$\mathcal{L}_r = w_{\mathrm{recon}}\mathcal{L}_{\mathrm{recon}} + w_{\mathrm{gan}}\mathcal{L}_{\mathrm{gan}} + w_{\mathrm{percep}}\mathcal{L}_{\mathrm{percep}}\,, \tag{5}$$

where $\mathcal{L}_{\mathrm{recon}}$ denotes the reconstruction loss, *i.e.*, the L1 distance between $\mathcal{I}^S_o$ and $\hat{\mathcal{I}}^S_o$. $\mathcal{L}_{\mathrm{gan}}$ is the generative adversarial network (GAN) loss, where we additionally train a discriminator to classify $\mathcal{I}^S_o$ and $\hat{\mathcal{I}}^S_o$. $\mathcal{L}_{\mathrm{percep}}$ means the perceptual loss, which is the L1 distance between the feature maps of $\mathcal{I}^S_o$ and $\hat{\mathcal{I}}^S_o$ in each level of VGGNet [41]. Even though the reconstruction is only supervised on training instances, we observe that it can roughly reconstruct novel views for unseen instances. We thus reason that the pre-trained voxel mapping can roughly encode the geometry of an instance.

**Detection base training** : In order to empower $\mathcal{M}(\cdot)$ with geometry encoding capability, we initialize it with the reconstruction pre-trained weights and conduct the instance detection training stage. In addition to the open-world detection loss [14], we introduce the instance classification loss $\mathcal{L}^{\mathrm{Ins}}_{\mathrm{cls}}$ and rotation estimation loss $\mathcal{L}^{\mathrm{Ins}}_{\mathrm{rot}}$ to supervise our VoxDet.

We define $\mathcal{L}^{\mathrm{Ins}}_{\mathrm{cls}}$ as the binary cross entropy loss between the true labels $\mathbf{s} \in \{0,1\}^N$ and the predicted scores $\hat{\mathbf{s}} \in \mathbb{R}^{N\times 2}$ from the QVM module. The rotation estimation loss is defined as:

$$\mathcal{L}^{\mathrm{Ins}}_{\mathrm{rot}} = \|\hat{\mathbf{R}}^Q\mathbf{R}^{Q\top} - \mathbf{I}\|\,, \tag{6}$$

Table 1: Overall performance comparison on synthectic-real datasets LM-O [17] and YCB-V [18]. Compared with various 2D methods, including correlation [5], attention [6], and feature matching [9, 19], our VoxDet holds superiority in both accuracy and efficiency. OLN* means the open-world object detector (OW Det.) [14] is jointly trained with the matching head while OLN denotes using fixed modules. $^{\dagger}$ the model is trained on both synthetic dataset OWID and real images.

| Test/Metric | | | LM-O | | | | YCB-V | | | | Avg. | | | |
| Method | OW Det. | Train | mAR | $AR_{50}$ | $AR_{75}$ | $AR_{95}$ | mAR | $AR_{50}$ | $AR_{75}$ | $AR_{95}$ | mAR | $AR_{50}$ | $AR_{75}$ | Speed |
|---|---|---|---|---|---|---|---|---|---|---|---|---|---|---|
| **VoxDet** | OLN* | OWID | **29.2** | **43.1** | **33.3** | 0.8 | **31.5** | **51.3** | **33.4** | 1.7 | **30.4** | **47.2** | **33.4** | 6.5 |
| $OLN_{Corr.}$ [14, 5] | OLN* | OWID | 22.3 | 34.4 | 24.7 | 0.5 | 24.8 | 41.1 | 26.1 | 0.7 | 23.6 | 37.8 | 25.4 | 5.5 |
| DTOID [6] | N/A | OWID | 9.8 | 28.9 | 3.7 | <0.1 | 16.3 | 48.8 | 4.2 | <0.1 | 13.1 | 38.9 | 4.0 | 2.8 |
| OS2D [7] | N/A | OWID | 0.2 | 0.7 | 0.1 | <0.1 | 5.2 | 18.3 | 1.9 | <0.1 | 2.7 | 9.5 | 1.0 | 5.3 |
| $OLN_{CLIP}$ [14, 19] | OLN | OWID$^{\dagger}$ | 16.2 | 32.1 | 15.3 | 0.5 | 10.7 | 25.4 | 7.3 | 0.2 | 13.5 | 28.8 | 11.3 | 2.8 |
| $OLN_{DINO}$ [14, 9] | OLN | OWID$^{\dagger}$ | 23.6 | 41.6 | 24.8 | 0.6 | 25.6 | 53.0 | 21.1 | 0.8 | 24.6 | 47.3 | 23.0 | 2.8 |
| Gen6D [5] | N/A | OWID$^{\dagger}$ | 12.0 | 29.8 | 6.6 | <0.1 | 12.1 | 37.1 | 5.2 | <0.1 | 12.1 | 33.5 | 5.9 | 1.3 |
| BHRL [42] | N/A | COCO | 14.1 | 21.0 | 15.7 | 0.5 | 31.8 | 47.0 | 34.8 | 1.4 | 23.0 | 34.0 | 25.3 | N/A |

Table 2: Overall performance comparison on the newly built real image dataset, RoboTools. For fairness, we only compare with the models fully trained on synthetic dataset here, more comparison see Appendix D. VoxDet shows superiority even under sim-to-real domain gap compared with other 2D representation-based methods [14, 5–7].

| Metric | OW Det. | mAR | $AR_{50}$ | $AR_{75}$ | $AR_{95}$ |
|---|---|---|---|---|---|
| **VoxDet** | **OLN*** | **18.7** | **23.6** | **20.5** | **5.1** |
| $OLN_{Corr.}$ [14, 5] | OLN* | 14.4 | 18.1 | 15.7 | 3.8 |
| DTOID [6] | N/A | 3.6 | 9.0 | 2.0 | <0.1 |
| OS2D [7] | N/A | 2.9 | 6.5 | 2.0 | <0.1 |

Table 3: Per module efficiency comparison. All the four methods share the same open-world detector [14]. Compared with 2D baselines that adopt cosine similarity [9, 19] or learnable correlation [5], our Voxel matching is more efficient, which shows $\sim \mathbf{2}\times$ faster speed. The numbers presented below are measured in seconds.

| Method/Module | Open-World Det. | Matching | ToTal |
|---|---|---|---|
| **VoxDet** | | **0.032** | **0.154** |
| $OLN_{CLIP}$ [14, 19] | | 0.248 | 0.370 |
| $OLN_{DINO}$ [14, 9] | 0.122 | 0.235 | 0.357 |
| $OLN_{Corr.}$ [14, 5] | | 0.060 | 0.182 |

where $\mathbf{R}^Q$ is the ground-truth rotation matrix of the query voxel. Note that here we only supervise the positive samples. Together, our instance detection loss is defined as:

$$\mathcal{L}_{\mathrm{d}} = w_1 \mathcal{L}_{\mathrm{cls}}^{\mathrm{RPN}} + w_2 \mathcal{L}_{\mathrm{reg}}^{\mathrm{RPN}} + w_3 \mathcal{L}_{\mathrm{cls}}^{\mathrm{Head}} + w_4 \mathcal{L}_{\mathrm{reg}}^{\mathrm{Head}} + w_5 \mathcal{L}_{\mathrm{cls}}^{\mathrm{Ins}} + w_6 \mathcal{L}_{\mathrm{rot}}^{\mathrm{Ins}}, \tag{7}$$

**Remark 1**: In both training stages, we only use the training objects, $\mathcal{O}_{\mathrm{base}}$. During inference, VoxDet doesn't need any further fine-tuning or optimization for $\mathcal{O}_{\mathrm{novel}}$.

## 4 Experiments

### 4.1 Implementation Details

Our research employs datasets composed of distinct training and test sets, adhering to $\mathcal{O}$base $\cap$ $\mathcal{O}$novel $= \phi$ to ensure no overlap between semantic classes of $\mathcal{O}$base and $\mathcal{O}$novel.

**Synthetic Training set:** In response to the scarcity of instance detection traing sets, we've compiled a comprehensive synthetic dataset using 9,901 objects from ShapeNet [15] and ABO [16]. Each instance is rendered into a 40-frame, object-centric $360^o$ video via Blenderproc [43]. We then generate a query scene using 8 to 15 randomly selected objects from the entire instance pool, each initialized with a random orientation. This process yielded 55,000 scenes with 180,000 boxes for training and an additional 500 images for evaluation, amounting to 9,800 and 101 instances respectively. We've termed this expansive training set "open-world instance detection" (OWID-10k), signifying our model's capacity to handle unseen instances. To our knowledge, this is the first of its kind.

**Synthetic-Real Test set:** We utilize two authoritative benchmarks for testing. LineMod-Occlusion [17] (LM-O) features 8 texture-less instances and 1,514 box annotations, with the primary difficulty being heavy object occlusion. The YCB-Video [18] (YCB-V) contains 21 instances and 4,125 target boxes, where the main challenge lies in the variance in instance pose. These datasets provide real test images while lacks the reference videos, we thus render synthetic videos using the CAD models in Blender.

**Fully-Real Test set:** To test the sim-to-real transfer capability of VoxDet, we introduced a more complex fully real-world benchmark, RoboTools, consisting of 20 instances, 9,109 annotations,

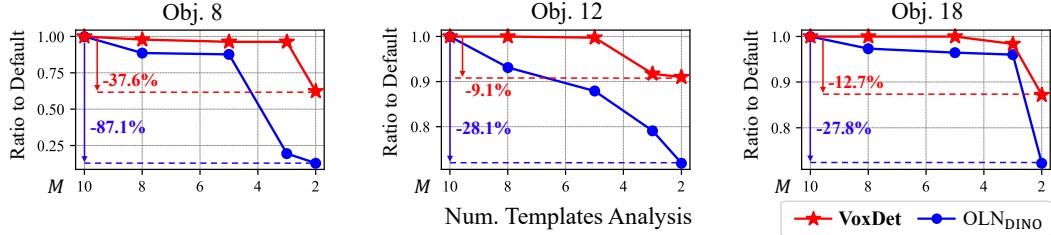

Figure 4: Number of templates analysis of VoxDet and 2D baseline, $OLN_{DINO}$ [14, 9] on YCB-V benchmark. Thanks to the learned geometry-aware 2D-3D mapping, VoxDet can work well with very few reference images, while 2D method suffers from such setting, dropping up to **87**%.

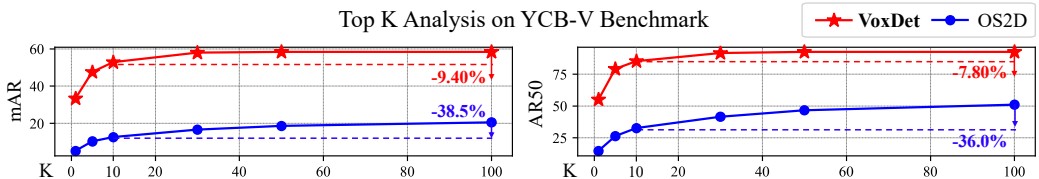

Figure 5: Top-K analysis of VoxDet and One-shot object detector [7]. By virtue of the instance-level matching method, QVM, VoxDet can better classify the proposals, so that $90\%$ of the true positives lie in Top-10, while for OS2D, this ratio is only $60\%$.

and 24 challenging scenarios. The instances and scenes are presented in Fig. 3. Compared with existing benchmarks [17, 18], RoboTools is much more challenging with more cluttered backgrounds and more severe pose variation. Besides, the reference videos of RoboTools are also real-images, including real lighting conditions like shadows. We also provide the ground-truth camera extrinsic.

**Baselines:** Our baselines comprise template-driven instance detection methods, such as correlation [5] and attention-based approaches [6]. However, these methods falter in cluttered scenes, like those in LM-O, YCB-V, and RoboTools. Therefore, we've self-constructed several 2D baselines, namely, $OLN_{DINO}$, $OLN_{CLIP}$, and $OLN_{Corr.}$ In these models, we initially obtain open-world 2D proposals via our open-world detection module [14]. We then employ different 2D matching methods to identify the proposal with the highest score. In $OLN_{DINO}$ and $OLN_{CLIP}$, we leverage robust features from pre-trained backbones [9, 19] and use cosine similarity for matching. [1] For $OLN_{Corr.}$, we designed a 2D matching head using correlation as suggested in [5]. These open-world detection based 2D baselines significantly outperform previous methods [5, 6]. In addition to these instance-specific methods, we also include a class-level one-shot detector, OS2D [7] and BHRL [42] for comparison.

**Hardware and configurations:** The reconstruction stage of VoxDet was trained on a single Nvidia V100 GPU over a period of 6 hours, while the detection training phase utilized four Nvidia V100 GPUs for a span of $\sim$40 hours. For the sake of fairness, we trained the methods referenced [5–7, 14, 19, 9] mainly on the OWID dataset, adhering to their official configuration. Inferences were conducted on a single V100 GPU to ensure fair efficiency comparison. During testing, we supplied each model with the same set of $M = 10$ template images per instance, and all methods employed the top $N = 500$ ranking proposals for matching. In the initial reconstruction training stage, VoxDet used 98% of all 9,901 instances in the OWID dataset. For each instance, a random set of $K = 4$ images were designated as output $\mathcal{I}_o^S$, while the remaining $M - K = 6$ images constituted the inputs $\mathcal{I}_i^S$. For additional configurations of VoxDet, please refer to Appendix A and our code.

**Metrics:** Given our assumption that only one desired instance is present in the query image, we default to selecting the Top-1 proposal as the predicted result. We report the average recall (AR) rate [44] across different IoU, such as mAR (IoU $\in 0.5 \sim 0.95$), $AR_{50}$ (IoU 0.5), $AR_{75}$ (IoU 0.75), and $AR_{95}$ (IoU 0.95). Note that the AR is equivalent to the average precision (AP) in our case.

## 4.2 Quantitative Results

**Overall Performance Comparison:** On the synthetic real datasets, we comprehensively compare with all the potential baselines, the results are detailed in Table 1, demonstrating that VoxDet consistently delivers superior performance across most settings. Notably, VoxDet surpasses the

---

[1] we default to the ViT-B model. DINO [9] and CLIP [19] might already be familiar with the test instances.

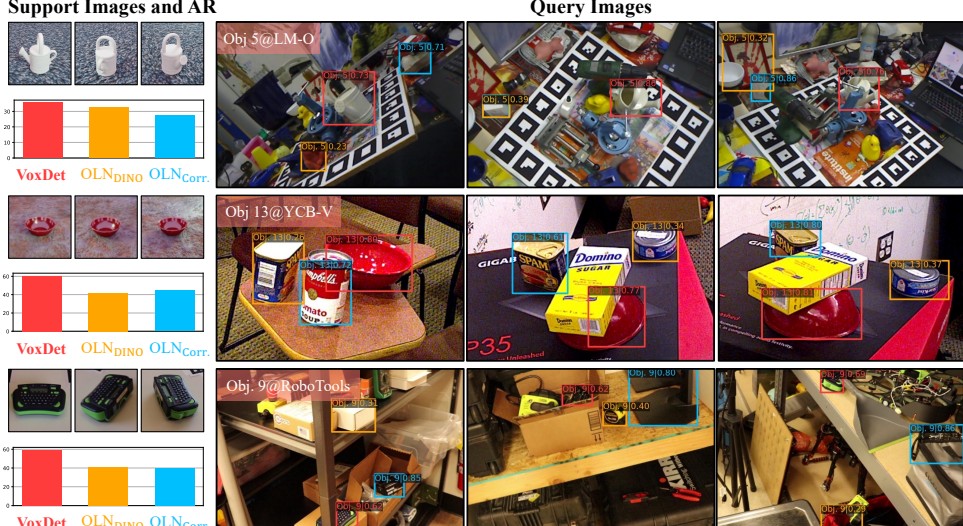

Figure 6: Detection qualitative results comparison between VoxDet and 2D baselines on the three benchmarks. VoxDet shows better robustness under pose variance (*e.g.* Obj. 5@LM-O first and second columns) and occlusion (*e.g.* Obj. 13@YCB-V second column and Obj. 9@RoboTools).

next best baseline, $OLN_{DINO}$, by an impressive margin of up to $20\%$ in terms of average mAR. Furthermore, due to its compact voxel representation, VoxDet is observed to be markedly more efficient. On the newly built fully real dataset, RoboTools, we only compare methods trained on the same synthetic dataset for fairness. As shown in Table 2, VoxDet demonstrates better sim2real transfering capability compared with the 2D methods due to its 3D voxel representation. We present the results comparison with the real-image trained models in Appendix D.

**Efficiency Comparison:** As QVM has a lower model complexity than $OLN_{CLIP}$ and $OLN_{DINO}$, it achieves faster inference speeds, as detailed in Table 3. Compared to correlation-based matching [5], VoxDet leverages the aggregation of multi-view templates into a single compact voxel, thereby eliminating the need for exhaustive 2D correlation and achieving $2\times$ faster speed.

In addition to inference speed, VoxDet also demonstrates greater efficiency regarding the number of templates. We tested the methods on the YCB-V dataset [18] using fewer templates than the default. As illustrated in Fig. 4, we found that the 2D baseline is highly sensitive to the number of provided references, which may plummet by $87\%$ when the number of templates is reduced from 10 to 2. However, such a degradation rate for VoxDet is $2\times$ less. We attribute this capability to the learned 2D-3D mapping, which can effectively incorporate 3D geometry with very few views.

**Top-K Analysis:** Compared to the category-level method [7], VoxDet produces considerably fewer false positives among its Top-10 candidates. As depicted in Fig. 5, we considered Top-$K = 1, 5, 10, 20, 30, 50, 100$ proposals and compared the corresponding AR between VoxDet and OS2D [7]. VoxDet's AR only declines by $5 \sim 10\%$ when $K$ decreases from 100 to 10, whereas OS2D's AR suffers a drop of up to $38\%$. This suggests that over $90\%$ of VoxDet's true positives are found among its Top-10 candidates, whereas this ratio is only around $60\%$ for OS2D.

**Ablation Studies:**

The results of our ablation studies are presented in Table 4. Initially, we attempted to utilize the 3D depth-wise convolution for matching (see the fourth row). However, this proved to be inferior to our proposed instance-level voxel relation. Reconstruction pre-training is crucial for VoxDet's ability to learn to encode the geometry of an instance (see the last row). Additionally, we conducted an ablation on the rotation measurement module (R) in the QVM, and also tried not supervising the predicted rotation. Both are inferior to our default settings.

Table 4: Ablation study for VoxDet in RoboTools benchmark. All the three critical modules are helpful in our design. Supervising the estimated rotation achieves slightly better results. Comparison with more matching module see Appendix B.

| Recon. | R | R w/ sup. | Voxel Rel. | mAR | $AR_{50}$ | $AR_{75}$ |
|--------|---|-----------|-----------|------|-----------|-----------|
| ✓ | ✓ | ✓ | ✓ | **18.7** | **23.6** | **20.5** |
| ✓ | ✓ | ✗ | ✓ | 18.2 | 23.2 | 20.0 |
| ✓ | ✗ | ✗ | ✓ | 15.6 | 21.9 | 17.0 |
| ✓ | ✓ | ✓ | ✗ | 15.1 | 19.4 | 16.2 |
| ✗ | ✓ | ✓ | ✓ | 14.2 | 18.3 | 15.7 |

Support     Query and Voxel Activation          Support     Query and Voxel Activation

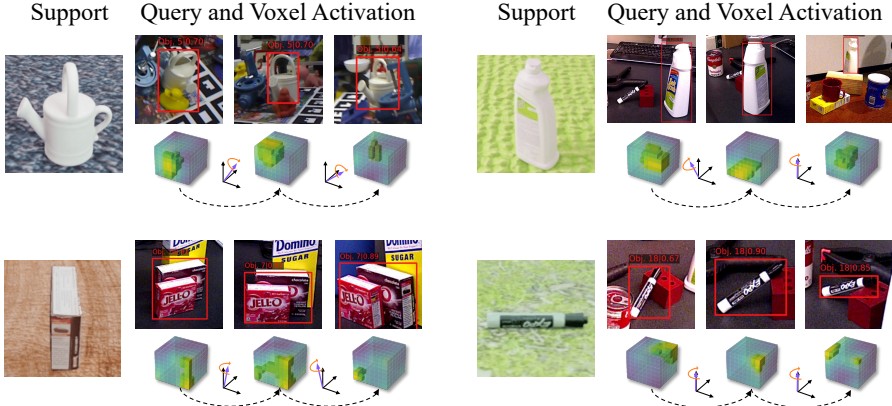

Figure 7: Visualization of the high activation grids during matching. As query instance rotates along a certain axis, the location of the high-activated grids roughly rotates in the corresponding direction.

## 4.3 Qualitative Results

**Detection Visualization** The qualitative comparison is depicted in Fig. 6, where we compare VoxDet with the two most robust baselines, OLNDINO and OLNCorr.. We notice that 2D methods can easily falter if the pose of an instance is not seen in the reference, e.g., 2-nd query image in the 1-st row, while VoxDet still accurately identifies it. Furthermore, 2D matching exhibits less robustness under occlusion, where the instance's appearance could significantly differ. VoxDet can effectively overcome these challenges thanks to its learned 3D geometry. More visualizations and qualitative comparisons see Appendix C.

**Deep Voxels Visualization** To better validate the geometry-awareness of our learned voxel representation, we present the deep visualization in Fig. 7. The gradient of the matching score is backpropagated to the template voxel and we visualze the activation value of each grid. Surprisingly, we discover that as the orientation of the query instance changes, the activated regions within our voxel representations accurately mirror the true rotation.

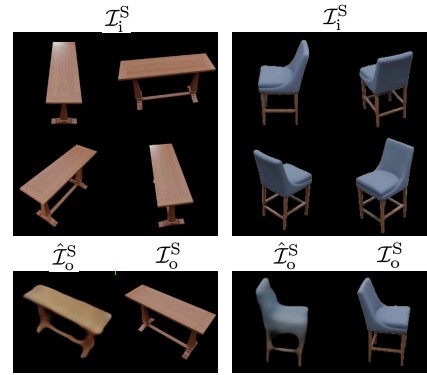

Figure 8: Reconstruct results of VoxDet on unseen instances. The voxel representation in VoxDet can be decoded with a relative rotation and synthesize novel views, which demonstrate the geometry embedded in our learned voxels.

This demonstrates that the voxel representation in VoxDet is aware of the orientation of the instance.

**Reconstruction Visualization** The voxel representation in VoxDet can be decoded to synthesize novel views, even for unseen instances, which is demonstrated in Fig. 8. The voxel, pre-trained on 9500 instances, is capable of approximately reconstructing the geometry of unseen instances.

## 5 Discussions

**Conclusion:** This work introduces VoxDet, a novel approach to detect novel instances using multi-view reference images. VoxDet is a pioneering 3D-aware framework that exhibits robustness to occlusions and pose variations. VoxDet's crucial contribution and insight stem from its geometry-aware Template Voxel Aggregation (TVA) module and an exhaustive Query Voxel Matching (QVM) specifically tailored for instances. Owing to the learned instance geometry in TVA and the meticulously designed matching in QVM, VoxDet significantly outperforms various 2D baselines and offers faster inference speed. Beyond methodological contributions, we also introduce the first instance detection training set, OWID, and a challenging RoboTools benchmark for future research.

**Limitations:** Despite its strengths, VoxDet has two potential limitations. Firstly, the model trained on the synthetic OWID dataset may exhibit a domain gap when applied to real-world scenarios, we present details in Appendix D. Secondly, we assume that the relative rotation matrixes and instance masks (box) for the reference images are known, which may not be straightforward to calculate. However, the TVA module in VoxDet doesn't require an extremely accurate rotation and 2D appearance. We present further experiments addressing these issues in Appendix E.

## Acknowledgement

This work was sponsored by SONY Corporation of America #1012409. This work used Bridges-2 at PSC through allocation cis220039p from the Advanced Cyberinfrastructure Coordination Ecosystem: Services & Support (ACCESS) program which is supported by NSF grants #2138259, #2138286, #2138307, #2137603, and #213296. The authors would also like to express the sincere gratitude on the developers of BlenderProc2 [43].

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

## Supplementary

To make our model fully reproducible, we present complete implementation details in Appendix A. Besides, our code library will be released upon acceptance. We report more comparisons between our QVM module and the 2D matching/relation techniques [1, 5, 45] in Appendix B to demonstrate the superiority of QVM in instance-level 3D matching. In Appendix C, we present more detection qualitative results. We further present some discussions about the sim2real domain gap of VoxDet in Appendix D. To test the robustness of VoxDet under interference on the voxel representation, we display results obtained from the flawed voxel in Appendix E. Finally, we provide extended related works discussions in Appendix F, where we exhaustively compare VoxDet with the existing instance-level tasks, including visual tracking, instance pose estimation, and instance retrieval.

## A  Implementation Details

**Model Structure:**  We adopt ResNet50 [46] with feature pyramid network [26] as our feature extractor $\psi(\cdot)$. The default multi-scale ROIAlign in [26] is leveraged to obtain the 2D proposal features, where the dimensions are $N = 500, C = 256, w = 7$. In our 2D-3D mapping, we set $C/d = 32, d = 8$, which results in the voxel feature dimension $C_v = 256, D = 16, L = 14$. All the 3D convolutions in TVA and QVM take kernel size as 3 and the padding equals to 1, so that the dimension of the voxels remains the same throughout the two modules. For the $\mathrm{Rot}(\cdot, \cdot)$ function, we have followed [10] to use `torch.nn.functional.affine_grid()` and `torch.nn.functional.grid_sample()` functionalities. Though the 2D-3D mapping can learn the rotations in the physical world, it sacrifices some semantics information in the feature channels when reshaping. Therefore, in QVM, we have a global matching branch to retrieve the lost semantic information. To be more specific, we apply global average pooling on the support features to get a support vector $\mathbf{k} \in \mathbb{R}^{1 \times C \times 1 \times 1}$. Then we adopt depth-wise convolution between $\mathbf{k}$ and $\mathbf{F}^Q$ to get a correlation map. Note that this correlation map preserved all the semantic channels from the backbone $\psi(\cot)$, so that the lost information in the 2D-3D mapping. The map is added to the voxel relation output $\mathcal{R}_v(\mathbf{V}^S, \mathrm{Rot}(\mathbf{V}^Q, \hat{\mathbf{R}}^Q))$ for the final score.

**Training Details:**  In the first reconstruction stage, we set the loss weights as $w_{\mathrm{recon}} = 10.0, w_{\mathrm{gan}} = 0.01, w_{\mathrm{percep}} = 1.0$. The model is trained for 16 epoch on the 9600 instances from OWID datasets. We leveraged Adam optimizer [47] with a base learning rate of $5 \times 10^{-5}$ during training. In the second detection stage, we initialize the 2D-3D mapping modules in TVA and QVM with the reconstruction pre-trained weights. VoxDet first only learns the detection task, without learning the rotation estimation, *i.e.*, the loss weights are set as $w_1 = w_2 = w_3 = w_4 = w_5 = 1.0, w_6 = 0$ in the first 10 epochs, where SGD is leveraged as an optimizer with 0.02 base learning rate. Note that in this stage, the 2D-3D mapping part only takes $\frac{1}{10}$ of the base learning rate. Then in the final epoch, VoxDet learns the rotation estimation with the detection part fixed, *i.e.*, $w_1 = w_2 = w_3 = w_4 = w_5 = 0.0, w_6 = 1.0$. However, supervising rotation is not the key requirements and is optional for VoxDet. It improves the performance slightly by $1 \sim 2\%$.

## B  More Matching Module Comparison

We compare QVM with more matching techniques in Table 5, where the averaged results onthe cluttered LM-O [17] and RoboTools benchmark are reported. We first ablate the Voxel Relation module in QVM, which results in QVM[†]. Specifically, all the Voxel Relation in QVM[†] are replaced by a simple depth-wise convolution, *i.e.*, we first apply global average pooling on the template voxel to get a feature vector, which is then taken as the convolution kernel to calculate the correlation voxel from the queries. We can see such a naive design will result in a performance drop.

Table 5: Comparison with different types of matching module. We compare QVM with the correlation in [5], class-level relation proposed in [1], and the class distance defined in FSDet [45].

| Method | mAR | AR$_{50}$ | AR$_{75}$ |
|---|---|---|---|
| **QVM (Ours)** | **23.95** | **33.35** | **26.90** |
| QVM[†] | 22.45 | 31.75 | 25.05 |
| 2D Relation [1] | 20.25 | 29.70 | 22.80 |
| FSDet [45] | 20.35 | 29.35 | 22.60 |
| Local Matching [48, 49] | 10.60 | 13.90 | 11.75 |

For all the rest methods, we used the same open-world detector to obtain the universal proposals, which are then matched with the template images using different matching techniques. To be more specific, 2D Corr. [5] constructs support vectors from every reference image. Then, depth-wise convolution is conducted between each support vector and the proposal patch. The resulting

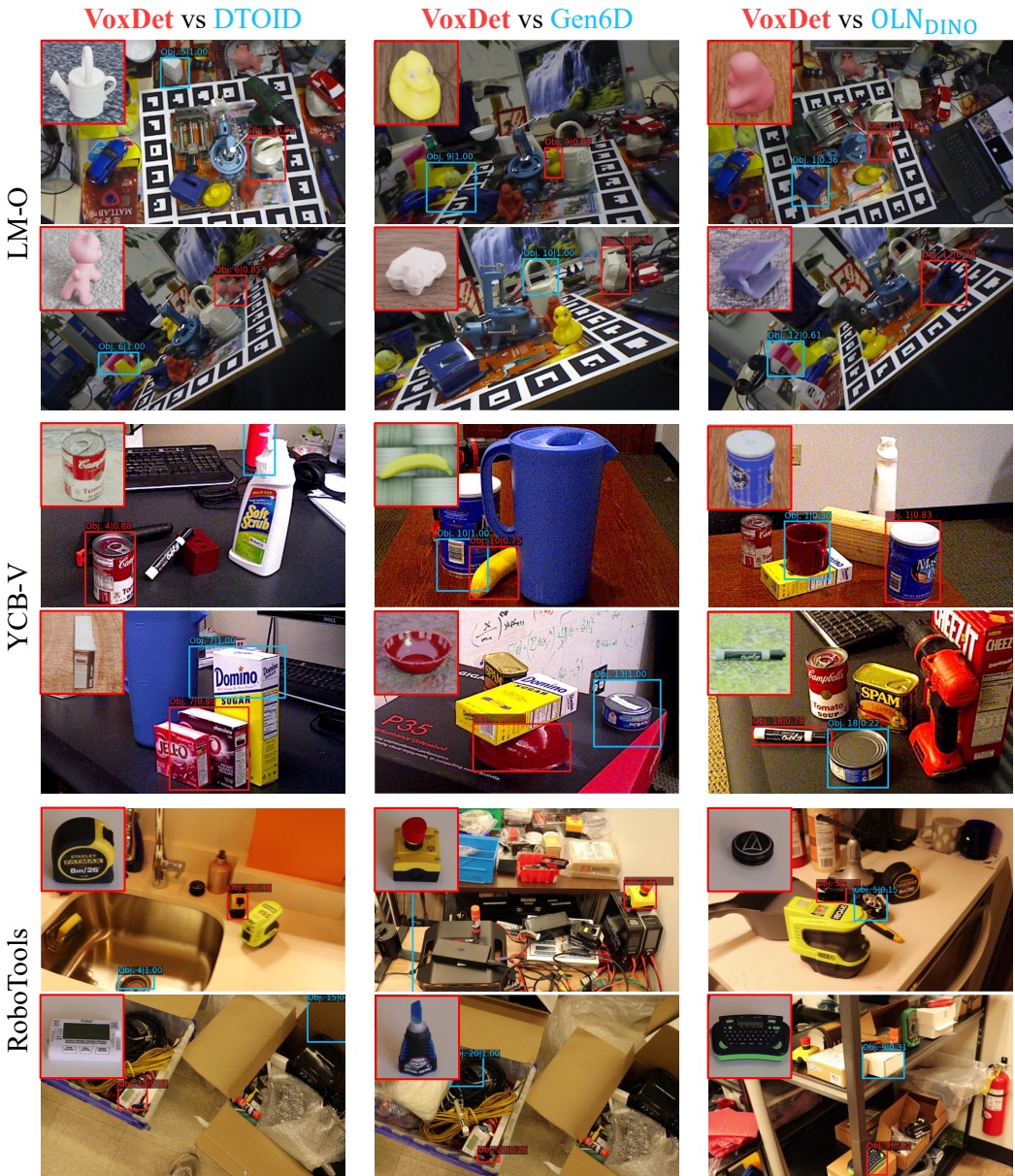

Figure 9: Detection qualitative results comparison between VoxDet and 2D baselines, DTOID [6], Gen6D [5], OLN$_{DINO}$ [14, 9] on the three benchmarks. VoxDet shows better robustness under pose variance and occlusion. These qualitative comparisons can be better visualized in our supplementary video.

correlation maps are sent to an MLP for classification score. In 2D Relation [1], we substitute the simple depth-wise convolution in 2D Corr. with the spatial and channel relation proposed in [1]. In FSDet [45], the depth-wise convolution in 2D Corr is replaced by the distance defined in [45]. Since they are geometry-unaware, we find all the 2D techniques worse than our QVM module.

Additionally, we designed a Local Matching baseline [49, 48]. In Local Matching, we first extract local key points from the reference images and proposals using SuperPoint [49]. Then the points descriptors are matched by SuperGlue [48]. We take the mean matching score of all the points in the proposal as their classification score. We find such an implementation, though geometry-invariant, falls short in our task since it lacks semantic representation of the whole instance.

## C  More Detection Visualizations

We present more detection qualitative comparisons in Fig. 9. VoxDet, in red, is compared with three baselines, DTOID [6], Gen6D [5], and $OLN_{DINO}$. Compared with previous instance detectors [6, 5], VoxDet is more robust under orientation variation and severe occlusion by virtue of the learned geometric knowledge. For example, in the LM-O benchmark, second column, when the duck is partially occluded and the egg box is in different orientations, VoxDet can still find them while Gen6D fails. Compared with similarity matching [9], VoxDet can better distinguish similar instances via the QVM module. For instance, in the RoboTools benchmark, the third column, the desired instance could be distracted by the motor, which has similar appearances but different geometry. Our VoxDet can discover such geometric differences and make correct classification, while the similarity matching falls short even if the feature from DINO [9] is stronger than ResNet50 [46].

## D  Sim-to-Real Comparison

VoxDet is entirely trained on synthetic dataset, OWID. We observe that the model shows some domain gap when transferred to real-world images like RoboTools. On the synthetic-real datasets, LM-O [17] and YCB-V [18], our model easily outperforms those trained on real images, while it shows limitations in fully real test set RoboTools. For example, Gen6D [5] is mainly trained on real-images, which reports 17.0 mAR, 35.5 $AR_{50}$, and 14.3 $AR_{75}$. Its $AR_{50}$ is higher than VoxDet (23.6) while in harder metrics like $AR_{75}$, our model works better (20.5). Compared with the cutting edge foundation models that are trained on large-scale real images, our model still shows spaces for improvement. For example, $OLN_{CLIP}$ achieves 11.0 mAR, 20.8 $AR_{50}$, and 9.2 $AR_{75}$, which is worse than VoxDet. Yet, $OLN_{DINO}$ [13] can outperform VoxDet in RoboTools with over 30 mAR. We conclude that the feature representation from the concurrent 2D foundation model [13] could be a stronger backbone for VoxDet to overcome the domain gap issue. Learning a geometry-aware strong voxel representation from such foundation model will be one of our future work.

## E  Performance under Flawed Voxel

VoxDet assumes known instance masks and poses for the reference video, which may have some noise during realworld deployment. To quantitatively analysis the robustness of VoxDet under flawed Voxels, we present its results on RoboTools when the reference video is disturbed in appearance and geometry.

Table 6: Performance of VoxDet on RoboTools when the reference is disturbed. The ratio means center shift and scale noise with respect to width and height.

| Mean Shift Ratio | 0 | 10% | 20% | 30% |
|---|---|---|---|---|
| $AR_{50}$ | 23.6 | 20.1 | 18.9 | 17.1 |

**Add noise on the reference image patches** :
We tried to add random shift on the cropped area in the reference images, resulting in inaccurate instance appearance. The results on RoboTools are shown in Table 6. We conclude that even when we disturb around 65% of the voxel (30% shift on each 2D patch), the model still works, which means VoxDet is robust to appearance noise.

**Add noise on the relative poses** : We tried to add random error on the pose of the reference images, resulting in inaccurate instance geometry. When we add as large as 15 degree angular error, the performance ($AR_{50}$) decreased from 23.6 to 20.4. We conclude that VoxDet is not very sensitive to the geometry noise.

## F  Extended Related Works

**Visual Object Tracking**  aims to localize a general target instance in a video, given its initial state in the first frame. Early methods adopt discriminative correlation filters [50–52], where the calculation in the frequency domain is so efficient that real-time speed can be achieved on a single CPU. More recently, methods are developed on Siamese Network [53] and Transformers [54–56]. Unlike detection, object tracking has a strong temporal consistency assumption, *i.e.*, the location and appearance of the instance in the next frame do not largely vary from the previous frame. So that they only conduct detection/matching in the small search region with a single 2D template, which can't work for our whole image detection setting.

**Instance Pose Estimation**  is developed to estimate the 6 DoF pose of an unseen instance. Some of them [57, 58] match the local point features and resort to RANSAC to optimize the relative pose. While others [5, 59] first selects the closest template frame and then conducts pose refinement on the

known template poses. Most of these methods usually assume the instance detection is perfect, *i.e.*, they crop the instance out of the query image with the ground truth box and estimate the pose on the small object-centered patch. Our VoxDet can serve as their front-end, which is robust to cluttered environments, thus making the detection-pose estimation framework more reliable.

**Instance Retrieval** hopes to retrieve a specific instance from a large database with a single reference image [60–65]. Some early work extracts local point features from template and query patch for image matching [61, 49], which may suffer from poor discriminative capability. More recent work resorts to the deep neural network for a global representation of the instance [62–65], which is compared with the features from query images. However, most of them construct 2D template features from the reference, so that their representation is unaware of the 3D geometry of the instance, which may not be robust under severe pose variation. Besides, instance retrieval methods usually require high-resolution query images for the discriminative features, while the instance in our cluttered query image could be in low-resolution, which sets additional barriers to these approaches.

