# OpenReview forum: "VoxDet: Voxel Learning for Novel Instance Detection"
_NeurIPS.cc/2023/Conference — NeurIPS 2023 spotlight_

### Official Review · Reviewer_NFHY · 2023-07-02

**Soundness:** 4 excellent
**Presentation:** 4 excellent
**Contribution:** 3 good
**Rating:** 7
**Confidence:** 3

**Summary:**

The paper introduces a new 3D geometry-aware framework, VoxDet, to address the challenge of detecting unseen instances based on multi-view templates. VoxDet includes three main modules: a Template Voxel Aggregation (TVA) module, an open-world detection module, and a Query Voxel Matching (QVM) module. TVA transforms multi-view 2D features to 3D template voxel feature and then accumulated according to pre-computed relative rotation. During detection, the detected 2D proposals are converted to query voxels through the learned 2D-3D mapping. QVM module estimates the relative rotation between the query and template, aligns them and performs a comparison.

Along with the introduction of this method, the authors also build a large-scale synthetic training dataset, Open-World Instance Detection (OWID), made from the ShapeNet and some objects datasets. Experiments on LineMod-Occlusion, YCB-video dataset.

**Strengths:**

1. As I am not directly working on this research topic, the main text give enough context knowledge to understand the overall method. The overall writing are smooth and easy to understand.
2. As claimed by the author, VoxDet is the first to employ the explicit 3D knowledge in the multi-view templates to perform open-world detection in the object-level household datasets.
3. In my view, the OWID dataset effectively complements existing household object datasets. It specifically addresses areas typically overlooked such as accommodating pose variations and occlusions of household objects, thereby enhancing our understanding and performance in these complex scenarios.
4. Experimental results show that the 3D geometry-aware VoxDet outperforms various previous works and different 2D baselines while achieving faster inference speed.


**Weaknesses:**

No obvious weakness in my view. See question section.

**Questions:**

The manuscript presents several intriguing concepts and experimental results. However, I have a few queries that need addressing for a more comprehensive understanding of the study.

1. While the paper extensively compares methods developed by the authors (as mentioned in lines 261-262), it raises the question of whether this is an entirely novel experimental setup. Would it be feasible to compare other open-world methods without necessarily using the same training data? This query stems from the desire to assess the actual impact of the proposed method across various techniques and its potential real-world applications.
2. I noticed that the supplementary material does not appear to include experiments testing robustness against accurately estimated rotation, as claimed in lines 341-343. Could you provide more details or point me towards this information?
3. I am curious whether the OWID and RoboTools benchmarks will be made publicly available to aid future research.
4. In line 192, the text mentions the experiment that benefits from the group kernel design. I am keen to understand why and how this group gets the advantage.

**Limitations:**

Yes.

---

> ### Author Rebuttal · Authors · 2023-08-09
>
> Dear Reviewer NFHY,
>
> Thanks for your insightful questions and feedback. We provide specific answers below:
>
> 1. **Is this a novel experiment setup?**
>
>      Thanks for this question.
>
>      Yes, this is a novel setup, we have developed new training data (OWID) and test benchmarks (authoritative data from LM-O, YCB-V, and newly built RoboTools) for the instance detection task. Gen6D, OS2D, DTOID, and $\mathrm{OLN}\_\mathrm{Corr.}$ are entirely trained on our OWID dataset.
>     For $\mathrm{OLN}\_\mathrm{CLIP, DINO}$, we use OWID to train the open-world detector and resort to an off-the-shelf pre-trained foundation model for matching.
>
> 2. **Would it be feasible to compare other open-world methods without necessarily using the same training data?**
>
>      Thank you for this question.
>
>      It is feasible.
>      * For Gen6D, OS2D, $\mathrm{OLN}\_\mathrm{Corr.}$, and DTOID, we have tried directly using their official model trained on their official 2D dataset, while it performs worse than re-training with our new OWID dataset, e.g., see the Gen6D results on YCB-v dataset as follows:
>
>           | Training set\Metric |  mAR   |   AR50   |  AR75   |
>           | :-----------------: | :----: | :------: | :-----: |
>           |        COCO         |  7.5   |   20.8   |   3.4   |
>           |      **OWID**       | **12** | **26.8** | **4.8** |
>
>         We believe OWID is a novel dataset focused on instance detection. We have tried our best to re-train the models to reported their highest score in the comparison.
>
>      * For $\mathrm{OLN}\_\mathrm{DINO}$ and $\mathrm{OLN}\_\mathrm{CLIP}$, we try our best to be fair by using the same open-world detector. Since we believe their pre-trained 2D representation from the large real-world dataset is good enough, we only used synthetic OWID to pre-train the open-world detection module and utilized the off-the-shelf CLIP and DINO models to match the reference images and proposals.
>
>      To summarize, we have tried our best to ensure fairness in all experiments. We conclude that VoxDet outperforms these baselines by virtue of the innovative incorporation of explicit 3D geometry. We will provide these details in the final version supplementary.
>
>
> 3. **More details about the robustness against inaccurate rotation estimation**
>
>      Thanks very much for raising this issue.
>
>      We tried to introduce random errors in the rotation angle of the reference poses. The results on RoboTools are shown below:
>
>      | Mean Angle Error (degree) |  0   |  5   |  10  |  15  |
>      | :-----------------------: | :--: | :--: | :--: | :--: |
>      |           AR50            | 14.0 | 13.6 | 13.1 | 12.7 |
>
>      We conclude that ***adding as large as a 15-degree error will not cause a significant performance drop***.
>
>      We will include these results in the final version supplementary. We'd like to thank you again for the careful inspection of our work.
>
> 4. **Will the dataset be publicly available?**
>
>     Yes, we promise to open source all the data, code, and models upon acceptance. (We have appended our code in the supplementary to ensure reproducibility)
>
>
> 5. **Detailed explanation of the group convolution**
>
>     Thanks for this technical question.
>
>     Please check L187-193 in our original submission, the combination of channel interleave and grouped convolution kernel is the key innovation of our QVM module, which effectively compares two voxels on the instance level.
>
>     Our key finding is that, when the model learns each corresponding channel independently, it better discovers the similarity and differences between two instance voxels.
>
>     To achieve this, we first interleave the feature channels of two voxels $\mathbf{v}_1,\mathbf{v}_2\in\mathbb{R}^{c\times a\times a\times a}$ so that the corresponding channels are "grouped" together as $\mathrm{In}(\mathbf{v}_1, \mathbf{v}_2)=[\mathbf{v}^1_1, \mathbf{v}^1_2, \mathbf{v}^2_1, \mathbf{v}^2_2, \cdots, \mathbf{v}^c_1, \mathbf{v}^c_2]\in \mathbb{R}^{2c\times a\times a\times a}$, where $\mathbf{v}^k_1, \mathbf{v}^k_2$ is the voxel feature in the $k$-th channel.
>     Then in the grouped convolution $\mathrm{Conv3D}(\mathrm{In}(\mathbf{v}_1, \mathbf{v}_2), \mathrm{group}=c)$, each kernel only learns to distinguish the same corresponding channels from the two voxels.
>
>     To validate this, we have tried using the simple standard 3D convolution to replace voxel relation, the averaged results on LM-O and YCB-v benchmarks are shown below:
>
>     |        Impementation         |   mAR    |   AR50   |   AR75   |
>     | :--------------------------: | :------: | :------: | :------: |
>     | **Voxel Relation (default)** | **21.7** | **25.4** | **19.1** |
>     |      Standard 3D Conv.       |   16.5   |   19.2   |   14.3   |
>
>     We will include more explanations and results in our final paper.

---

> > ### Comment · Reviewer_NFHY · 2023-08-14
> > **Thanks for the authors' rebuttal**
> >
> > The author's rebuttal has addressed my concern. I raised the score to accept.

---

### Official Review · Reviewer_dtGe · 2023-07-04

**Soundness:** 4 excellent
**Presentation:** 4 excellent
**Contribution:** 3 good
**Rating:** 7
**Confidence:** 4

**Summary:**

This paper tackles the task of novel instance detection when given multi-view reference images.
The key is the utilization of a 3D voxel representation, contrary to 2D representations used by prior work that fair under occlusion and pose variation.
The proposed detector, VoxDet, combines reference features into a 3D template voxel through the Template Voxel Aggregation module and compares that to voxels similar generated from object proposals generated from a RPN through the Query Voxel Matching module.
By aligning and comparing the resulting voxels, the detections are more robust to pose variations than existing methods.
Extensive experimental results are shown on existing benchmarks as well as the newly proposed RoboTools benchmark, which contains much more clustered scenes to test existing methods.
Furthermore, a synthetic training dataset, Open-World Instance Detection, is also introduced, which contains numerous training examples sourced from object datasets.

**Strengths:**

- The underlying motivation of the method makes intuitive sense. By leveraging a 3D representation of every object, it is much more robust to changes in viewpoint, which 2D methods struggle with.
- The proposed method achieves significantly better performance compared to previous methods across all benchmarks, even when compared to their own strong baselines.
  - Qualitatively, the provided visualizations and the video in the supplementary clearly demonstrates the improvement in robustness in detection.
- The paper also introduces both a large-scale training dataset and a new evaluation benchmark, both of which are good contributions to the field of instance detection.
- The paper is very well written and easy to understand.

**Weaknesses:**

- The training process requires rotation to be known for the given reference images, which may not be available and are also not used by other methods.
  - The results in Table 3 seems to suggest that without supervising rotation, the performance drops by 1 point in mAR, which is quite significant (though still higher than other methods).
  - Can the authors explain the "R" and "R w/ sup" column in Table 3? This is not well explained.
- Comparison to 2D baselines:
  - Are the previous 2D methods trained with the open-world detector used by the proposed method?
  - What is the pre-training procedure of these baselines on the OWID dataset? Comparison can be made without pre-training to compare the performance without the OWID dataset.
  - The paper lacks comparisons to more recent few-shot/one-shot detection methods, such as [1, 2, 3].

[1] Yang, Hanqing, et al. "Balanced and hierarchical relation learning for one-shot object detection." Proceedings of the IEEE/CVF Conference on Computer Vision and Pattern Recognition. 2022.
[2] Qiao, Limeng, et al. "Defrcn: Decoupled faster r-cnn for few-shot object detection." Proceedings of the IEEE/CVF International Conference on Computer Vision. 2021.
[3] Zhang, Gongjie, et al. "Meta-DETR: Image-level few-shot object detection with inter-class correlation exploitation." arXiv preprint arXiv:2103.11731 (2021).

**Questions:**

See weaknesses section.

Writing:
- Line 197: Typo "haed" -> "head".
- Line 246: Typo "og" -> "of".

**Limitations:**

The paper discusses several limitations of the proposed method.

---

> ### Author Rebuttal · Authors · 2023-08-09
>
> Dear Reviewer dtGe,
>
> Thank you very much for the questions and comments. We address them as follows:
>
> 1. **VoxDet requires known rotations of the references during training.**
>
>      Thanks for raising this concern.
>
>      * It is true that most 2D methods didn't consider the geometry information during training. Our intuition is that humans understand/remember one specific instance not only through its appearance but also via its geometry. The geometry and shape can help the search task under occlusion or pose variation challenges. Motivated by this, we have incorporated explicit geometry to construct a compact instance voxel, which enables more effective and efficient detection.
>
>      * We would like to mention that ***in OWID synthetic dataset, the geometry information is readily available to make the system better***, while other 2D methods can't cooperate with such information due to their network design.
>
>      * ***Durining inference, VoxDet doesn't need perfect reference poses***. We tried to introduce errors in rotation angle, which only results in a 3-9% performance drop:
>        | Mean Angle Error (degree) |  0   |  5   |  10  |  15  |
>        | :-----------------------: | :--: | :--: | :--: | :--: |
>        |           AR50            | 14.0 | 13.6 | 13.1 | 12.7 |
>        |           Drop            |  0   | 2.8% | 6.4% | 9.3% |
>
>      * Moreover, we could resort to any off-the-shelf SLAM/SfM system to obtain the reference poses during inference.
>
> 2. **More Explanations to Table 3**
>
>      Thanks for raising this up.
>
>      - In Table 3, "R" means "measuring query rotation in QVM w/o supervision", and "R w/ supervision" means "measuring query rotation in QVM w/ GT supervision".
>
>      - From Table 3, we have the following conclusions
>        1. ***Though w/o supervision, the structure design "Measuring R" in QVM can improve the performance of VoxDet, see last and second last row, AR50 and AR75***. Intuitively, when we match the instance in 3D, we would first roughly estimate its relative pose to better compare the appearance. While such improvements are not very robust in simpler metrics (mAR, which means IoU thresh from 0.25 to 0.95).
>        1. ***"Rotation supervision" can further improve the robustness of QVM, see the last and the first row.*** We by default used supervision since the ground truth is readily available in our synthetic dataset. And the result suggests that supervising this intermediate value (rotation) is better since we give the model more guidance for performing the appropriate transform. Yet, without supervision, our system still achieves SOTA performance.
>
>
>       We will carefully polish our writing and table captions in our final version to avoid confusion. We'd also like to answer any further questions or discussions.
>
> 3. **Details about the 2D baselines**
>
>      - $\mathrm{OLN}_\mathrm{CLIP, DINO, Corr.}$ are our self-crafted baselines, ***they all used the same open-world detection model***. For the other baselines, we have used the official code library and model structure without modification.
>
>      - For the baselines $\mathrm{OLN}_\mathrm{CLIP, DINO}$, we pre-train the open-world detection module and directly used the off-the-shelf foundation models' feature for cosine-similarity-based matching. We believe that the large-scale real-image base training has enabled strong representation from the foundation models. So that they are not tuned on OWID synthetic dataset in our experiments.
>
>      - The others are entirely trained on OWID for fairness, we have followed their official code and tried our best to reproduce the highest score. We also tried directly using their official pre-training, but the results are worse. For example, the results for Gen6D are shown below:
>
>        | Training set\Metric |  mAR   |   AR50   |  AR75   |
>        | :-----------------: | :----: | :------: | :-----: |
>        |        COCO         |  7.5   |   20.8   |   3.4   |
>        |      **OWID**       | **12** | **26.8** | **4.8** |
>
>      We will provide this detailed information in the final version of our supplementary.
>
> 4. **Comparison to more recent few/one-shot detectors**
>
>    Thank you for this constructive suggestion.
>
>    - We agree to include BHRL and Meta-DETR in the experimental comparison since they share a similar pipeline (base training + inference) and siamese network structure with VoxDet.
>
>      The comparison on the LM-O dataset is shown below, where ***VoxDet outperforms these class-level detectors by a larger margin***.
>
>
>      | Model |  VoxDet  | BHRL | Meta-DETR |
>      | :---: | :------: | :--: | :-------: |
>      |  mAR  | **27.4** | 14.1 |   12.0    |
>      | AR50  | **36.9** | 22.8 |   15.3    |
>      | AR75  | **26.4** | 14.6 |   11.6    |
>
>    DeFRCN is a transfer-learning-based few-shot detector, it requires exhaustive fine-tuning on the few reference images, which is inefficient for instance-search tasks. Besides, we find it hard (and unfair) to adjust the fine-tuning hyper-parameters on our new dataset. We thus didn't consider it in our experiments.
>
>    We will include BHRL and Meta-DETR in our final version.

---

> > ### Comment · Reviewer_dtGe · 2023-08-14
> > **Follow-up by Reviewer**
> >
> > I thank the authors for the detailed responses to my concerns, they have been adequately addressed. I have read through all the other reviews and the rebuttal, and I will maintain my score of "Accept".

---

### Official Review · Reviewer_W6QG · 2023-07-07

**Soundness:** 3 good
**Presentation:** 3 good
**Contribution:** 3 good
**Rating:** 7
**Confidence:** 3

**Summary:**

The manuscript introduces VoxDet, a system for novel instance detection. The method takes in a set of query images of an object instance and aims to detect this instance in a target image. The core novelty of the approach lies in the lifting of the query instance and all potential target instances (detected via an open world 2d detector) into a 3D voxel feature grid. Matching between the query voxels and the proposal voxels is carried in 3D instead of 2D (which is the current standard approach). The manuscript that listing the comparison into 3D yields superior novel instance detection performance to staying in 2D.


**Strengths:**

The paper approaches the hard task of novel instance detection from the interesting novel perspective of lifting the instance observations into 3D before the scoring and retrieval. The key intuition is that a fused 3d representation (than also has learned 3D object priors via the reconstruction objective) should be better at detection than pure 2D approaches which are current state of the art. The paper convincingly shows that this intuition holds using ablation studies and comparison to various 2D baselines (including using some of the cutting edge foundational models DINO and CLIP).

The paper uses high quality visuals to convey the core ideas and qualitative results.
The authors promise to release a novel dataset (OWID) and benchmark (RoboTools).




**Weaknesses:**

To me the main weakness is the clarity of the paper in the writing and the figures showing the system. I would not trust myself to be able to reimplement this syste:
While the figures and visuals (1 and 2) are high quality, I did have a hard time following them - a lot is going on!
- Fig 1: I was initially confused as to what is being Reconstructed or aggregated (its not clear from the figure)
- Fig 2: Even after reading the text and (hopefully having understood the system) this figure is hard to parse for me.
The writing was also in parts hard to follow for me. For example the Voxel Relation paragraphs - It seems voxel relation is about rotating voxels into the same coordinate system to be able to compare and score them?

Part of the contribution of the paper is a novel dataset of rendered objects, as far as I can tell. I would want to see some example renderings of this dataset to be able to judge the quality and potential impact of it.


**Questions:**

see weaknesses

**Limitations:**

Some limitations are addressed by the authors (sim-to-real gap and rotation for query images). Another important one that I was wondering about is the fact that the training dataset contains 10k object instances. From the writing its not clear how diverse those instances are and hence how much of the space of all possible objects we can expect this to cover?

---

> ### Author Rebuttal · Authors · 2023-08-09
>
> Dear Reviewer W6QG,
>
> We sincerely appreciate your feedback and concerns.
> We provided detailed explanations and responses as follows:
>
> 1. **Concerns to reimplement this system due to complex framework figures**
>
>      We sincerely appreciate this concern.
>
>      - To ensure reproducibility and help readers better understand our system, ***we have provided our code library in the supplementary material during initial submission.***
>
>      - We promise to release all the code, pre-trained models, and data upon acceptance.
>
> 2. **Figures 1 and 2 are not very clear**
>
>      Thank you for pointing this out.
>
>      * Figure 1: We first resort to the reconstruction objective to pre-train the 2D-3D mapping in the Voxel Aggregation module. The aggregation module is then partially initialized from the reconstruction of pre-trained weights.
>      * Figure 2: The first voxel relation is used to estimate query rotation, which is then utilized to rotate voxels into the same coordinate system. The second voxel relation is to compare and score the rotated voxels.
>
>      Following your valuable advice, we will carefully polish Figures 1 and 2 with more detailed explanations to avoid confusion. In the final version, we'd like to polish our presentation with any further suggestions to help the readers understand our system.
>
>
> 3. **Better to display some example renderings from OWID**
>
>    We express our sincere gratitude for this constructive suggestion.
>
>    ***We present a figure in the Rebuttal PDF, displaying representative reference images and test scenes in the OWID dataset.*** We will add this figure to the final version of our supplementary and include a more detailed explanation of the synthetic dataset.
>
> 4. **How diverse is OWID?**
>
>    Thanks for this question.
>
>    * The 10k instances in OWID are sampled from the combination of ShapeNet and ABO datasets. Specifically, for each class in ShapeNet (47 classes excluding the classes appearing in the test), we randomly sampled 60 CAD models, resulting in a total of around 3k instances. We used all of the 8k available instances in the ABO dataset. These data together contribute to the exact number of instances in OWID, 10,860. We will follow your kind advice and provide these details in our final supplementary.
>
>    * ***From our qualitative and quantitative experiments, VoxDet can generalize well to unseen instances (w/ unseen class label) using these data.***
>
>    * In the future, we are also planning to incorporate the most recent large-scale 3D CAD model dataset, *e.g.*, objaverse or OmniObject3D, for a larger instance-level training set.

---

> > ### Comment · Reviewer_W6QG · 2023-08-14
> > **response**
> >
> > Thanks for the response and clarification. Publishing the code should make the method reproducible. The added figures in the pdf for the OVID dataset are very useful to clarify the dataset. Thank you. Looks great with a lot of diversity.
> > I will upgrade to accept based on this.

---

### Official Review · Reviewer_6hBZ · 2023-07-07

**Soundness:** 3 good
**Presentation:** 3 good
**Contribution:** 3 good
**Rating:** 6
**Confidence:** 3

**Summary:**

The paper introduces a method to address the problem of novel instance detection. The key challenge of this problem over object detection is that the network needs to detect novel types of objects that were not in the training data. In contrast to prior art, this paper presents  a 3D geometry-aware framework that is built upon a 3D volume representation. Given a few images of the target instance, It first uses a template voxel aggregation module to transform multi-view information into 3D voxel features. The Query Voxel Matching module then aligns 2D queries with the template voxel. The proposed method outperforms various 2D baselines on LineMod-Occlusion, YCB-video, and RoboTools benchmarks.

**Strengths:**

- The proposed 3D geometry aware representation has cleared outperform the 2D counterpart by a large margin.

- This paper introduces a synthetic and a real-world datasets or training and evaluation for novel instance detection respectively. The research community can benefit from this new benchmark.

**Weaknesses:**

- When aligning 3D features of the support templates, why not consider both translation and rotation?

- The authors should make it clear that the proposed method requires posed images, where the camera poses are estimated by an external module, while previous methods use unposed images.

- What is the minimum number of support templates required by the proposed method in the inference stage? It would help readers to understand the performance of the proposed method better if the authors can provide an analysis on how the performance change as the number of support templates increases.

**Questions:**

Please refer to the weaknesses section

**Limitations:**

Yes

---

> ### Author Rebuttal · Authors · 2023-08-09
>
> Dear Reviewer 6hBZ,
>
> Thanks a lot for providing valuable suggestions and raising insightful questions about our work.
> We address and answer your questions below:
>
> 1. **Why not consider both translation and rotation in the alignment?**
>
>    Thanks for raising up this issue.
>
>    ***In the TVA module, we assume the cropped image patches are object-centered with little translation***. The reference video is assumed to be rotation-dominant by task setting. Small translations between reference frames can be compensated by the provided rotation. We thus intuitively didn't consider the translation in the template voxel alignment.
>
>    ***In the QVM module, we have tried to estimate both translation and rotation before alignment***. However, we observe that additionally estimating query translation may not benefit detection tasks. The results on the YCB-v benchmark are shown below:
>
>    | Translation Estimation |   mAR    |   AR50   |   AR75   |
>    | :--------------------: | :------: | :------: | :------: |
>    |    **No (default)**    | **33.3** | **55.0** | **33.4** |
>    |          Yes           |   29.4   |   47.6   |   32.1   |
>
>    This could be because:
>     * Training set has different translation scales, which makes learning the translation hard.
>     * Translation scales in training may be different from testing, which sets a barrier to a generalizable translation estimation module.
>
>     We will include more intuitive explanations in our final version supplementary. Thank you again for this constructive feedback.
>
> 2. **VoxDet requires posed images, while previous methods use unposed images**
>
>     Thanks for this concern.
>
>     * We agree that VoxDet introduced explicit geometry for the detection task, which is different from the previous 2D methods.
>
>     * ***VoxDet doesn't require perfect reference poses***. We tried to introduce noisy estimation on the rotation angle, which only results in a 3-9% performance drop:
>        | Mean Angle Error (degree) |  0   |  5   |  10  |  15  |
>        | :-----------------------: | :--: | :--: | :--: | :--: |
>        |           AR50            | 14.0 | 13.6 | 13.1 | 12.7 |
>        |           Drop            |  0   | 2.8% | 6.4% | 9.3% |
>
>     * In practice, with the current advancement of SLAM/SfM systems, having posed images become more feasible.
>
>     We will carefully polish the writing in the paper to avoid confusion.
>
>
> 3. **What's the minimum number of support templates required by VoxDet?**
>
>    Thanks for this question.
>
>    * Please kindly refer to L264 in the original submission. ***By default, we uniformly sampled 10 images*** from the 360-degree object-centric video.
>
>    * We also conducted additional experiments, proving fewer images for VoxDet. The results on RoboTools are shown below:
>
>        | Num. Ref | 10 (default) |  8   |  6   |  4   |  3   |  2   |  1   |
>        | :------: | :----------: | :--: | :--: | :--: | :--: | :--: | :--: |
>        |   mAR    |   **16.1**   | 16.2 | 16.0 | 15.9 | 16.0 | 14.9 | 12.3 |
>        |   AR50   |   **14.0**   | 13.9 | 13.9 | 13.7 | 13.7 | 12.9 | 11.2 |
>        |   AR75   |   **11.8**   | 11.9 | 11.7 | 11.7 | 11.8 | 11.0 | 9.3  |
>
>        We conclude that ***given only 3 (uniformly sampled) reference images, VoxDet can still work without obvious performance drop***. We attribute this *template efficiency* to the incorporated explicit geometric information and the implicitly learned 2D-3D mapping. We will add these additional results in the supplementary.
>    * Moreover, it is worth mentioning that, ***unlike VoxDet, the 2D baselines may suffer from fewer reference***. Please see Figure 3 in the paper for more details.
>
>    We will provide these additional experiments and analyses in our final version.

---

> > ### Comment · Reviewer_6hBZ · 2023-08-14
> >
> > Thanks the authors for providing feedback to my questions. They have resolved my concerns on the technical side of the paper so I remain positive about the paper. However, please make it clear in the revised version that the posed images are needed by the framework so readers understand the scope of this work better.

---

### Official Review · Reviewer_7Hv8 · 2023-07-08

**Soundness:** 3 good
**Presentation:** 3 good
**Contribution:** 3 good
**Rating:** 7
**Confidence:** 3

**Summary:**

The paper presents a new 3D geometry-aware instance detection framework, VoxDet, which effectively transforms multi-view 2D images into 3D voxel features using a Template Voxel Aggregation (TVA) module. This representation demonstrates improved resilience to occlusions and pose variations. VoxDet also uses a Query Voxel Matching (QVM) module to estimate relative rotation and compare aligned voxels for instance detection. The proposed framework demonstrates superior performance on various benchmarks including LineMod-Occlusion, YCB-video, and the newly constructed RoboTools.

**Strengths:**

+ VoxDet represents a significant shift from 2D to 3D instance detection, which could potentially handle more complex detection scenarios, particularly in cases of occlusions and varying object poses.
+ The creation of the OWID dataset and RoboTools benchmark contributes to the research community, providing valuable resources for future work in instance detection.

**Weaknesses:**

1. The training of VoxDet on synthetic data raises concerns about a potential domain gap when deployed in real-world situations. Real-world conditions such as varied lighting, noise, and complex backgrounds might not be adequately represented in synthetic datasets.
2. VoxDet's performance is contingent on the accurate synthesis of the 3D voxel by the TVA module. Any inaccuracies in the voxel representation could impact the detection process, potentially leading to incorrect identification or missed detection of instances.

**Questions:**

1. It is unclear how effectively VoxDet can detect texture-less objects or objects that have similar appearances but different geometric structures
2. How easily can VoxDet be adapted to detect instances from different object classes? Generalizability is key in practical applications where objects to be detected may span a wide variety.

**Limitations:**

A limitation is the model's assumption of the availability of a known relative rotation matrix for reference images. This could be hard to procure in many real-world scenarios.

---

> ### Author Rebuttal · Authors · 2023-08-09
>
> Dear Reviewer 7Hv8,
>
> We would like to express our sincere gratitude for your careful inspection and constructive feedback on our paper.
> We address your specific concerns as follows:
>
> 1. **Synthetic training data raises concerns about a potential domain gap**
>
>    Thanks for this concern.
>
>    * ***We have compared with the models trained with much larger real-world images*** (which is acknowledged by Reviewer W6QG), *i.e.*, $\mathrm{OLN}_\mathrm{DINO, CLIP}$. Though the 2D representations from CLIP and DINO are learned from large-scale high-quality real-world images, VoxDet outperforms them in instance detection due to our innovative and carefully designed voxel learning paradigm.
>    * Besides, ***We have built one real-world dataset*** (RoboTools) with complicated background, realistic lighting, and noise. Our experiments show that among all the methods that are trained with synthetic data, our approach is the least affected by the sim-to-real gap. ***We have added more visualization results under real-world challenges in the Rebuttal PDF for better understanding***.
>
>    * In addition to VoxDet, we also found that ***other 2D models trained with OWID actually outperforms the ones trained with real-world dataset***. We tried to train Gen6D with COCO and OWID, and the results on YCB-v is listed below.
>
>        | Training set\Metric |  mAR   |   AR50   |  AR75   |
>        | :-----------------: | :----: | :------: | :-----: |
>        |        COCO         |  7.5   |   20.8   |   3.4   |
>        |      **OWID**       | **12** | **26.8** | **4.8** |
>
>      For fairness, we keep the scale of both training set similar (both have ~200k training samples). We conclude that OWID provides instance-tailored data, which compensates for its synthetic nature and finally brings better results in instance detection task. ***We also display some rendering examples in the Rebuttal PDF, where we believe OWID included common real-world challenges.***
>
> 2. **Any inaccurate template voxel  may cause detection failure**
>
>    We sincerely appreciate this insightful concern.
>
>    To test the robustness of VoxDet under inaccurate template voxel, we further conducted the following experiments:
>
>    - ***Add noise on the reference image patches***
>
>      We tried to add random shift on the cropped area in the reference images, resulting in inaccurate instance appearance. The results on RoboTools are shown below:
>
>      | Mean Shift Ratio |  0   | 10%  | 20%  | 30%  |
>      | :--------------: | :--: | :--: | :--: | :--: |
>      |       AR50       | 14.0 | 13.7 | 12.8 | 11.5 |
>
>      The ratio means center shift and scale noise with respect to width and height. We conclude that ***even when we disturb around 65% of the voxel (30% shift on each 2D patch), the model still works***, which means VoxDet is robust to appearance noise.
>
>    - ***Add noise on relative poses***
>
>      We tried to add random error on the pose of the reference images, resulting in inaccurate instance geometry. The results on RoboTools are shown below:
>
>      | Mean Angle Error (degree) |  0   |  5   |  10  |  15  |
>      | :-----------------------: | :--: | :--: | :--: | :--: |
>      |           AR50            | 14.0 | 13.6 | 13.1 | 12.7 |
>
>      0 means without noise, from 1 to 3, we introduce larger and larger angular error (5 to 15 mean error). We conclude that ***when we disturb 15 degree of the reference poses (which is a very large error in pose estimation), the model drops by only 9%***, which means VoxDet is robust to geometry noise.
>
>    We will add these experiments and analysis in the final version of our appendix. Thanks again for your constructive feedback.
>
> 3. **Effectiveness on textureless and similar objects**
>
>    Thanks for this question.
>
>    * ***We have included LM-O as one of our test sets, which includes 8 textureless objects (see L240-241)***. VoxDet outperforms other 2D baselines and shows better effectiveness in these textureless objects. See Table 1 for quantitive results and Figure 5, supplementary video, and Rebuttal PDF Figure 2 for qualitative visualizations.
>
>    * ***We tried to introduce similar objects distractions for tes.***, We have displayed some samples in Rebuttal PDF Figure 2, the first row. Under this distraction, our model still works well.
>
> 4. **How easily can VoxDet be adapted to detect instances in different classes?**
>
>    We sincerely appreciate this question.
>
>    * In the OWID training set ***we have excluded all the possible overlapping classes during test***, so that the test instances are all unseen classes.
>
>    * Besides, ***VoxDet doesn't need any further tuning or optimization when deployed on a novel instance***.
>
>    Therefore, we believe VoxDet can generalize to unseen instances (classes) effectively and efficiently.
>
> 5. **VoxDet has the assumption that the relative poses of template images are known**
>
>    Thanks for pointing this out.
>
>    * VoxDet is robust to inaccurate poses of the reference images. We tried introducing a 15-degree error on the rotation angle, which only results in a 9% performance drop as shown above.
>
>    * In practice, we can resort to any off-the-shelf SLAM/SfM external modules, making the achievement of the poses easy.

---

### Author Rebuttal · Authors · 2023-08-09

Dear AC, Reviewers,

We would like to express our sincere gratitude for the careful inspection and constructive feedback from all the reviewers and the area chair. We are glad to see that **all of the reviewers in general hold a positive attitude** towards our paper in the pre-rebuttal period.

For positive comments, ***significant shift from pure 2D to 3D-aware detection*** (7Hv8, W6QG, NFHY), ***reasonable motivation and intuition*** (dtGe, W6QG), ***significantly better performance*** (6hBZ, W6QG, dtGe, NFHY), ***well-written pape***r (dtGe, NFHY), and ***favorable OWID and RoboTools datasets*** (All), we appreciate them and will carry them forward in our future work.

For negative comments, we first summarize and address the most general concern here:

1. **VoxDet assumes posed reference images**

    * We agree that different from previous 2D methods, VoxDet incorporates explicit 3D geometric information for instance representation and matching. We believe this is one of the innovations of our system.

    * Yet, we also hope to mention that ***VoxDet is robust to inaccurate poses of the reference images***. We tried to add as large as a 15-degree mean error on the rotation angle, which only results in a 9% performance drop as shown below:

       | Mean Angle Error (degree) |  0   |  5   |  10  |  15  |
       | :-----------------------: | :--: | :--: | :--: | :--: |
       |           AR50            | 14.0 | 13.6 | 13.1 | 12.7 |
       |           Drop            |  0   | 2.8% | 6.4% | 9.3% |

    * In practice, we can resort to SLAM/SfM systems (which are already embedded in many smart phones), making the availability of the poses easy.

    We will include the additional experiments in our final version and better polish our writing to avoid confusion.

1. **Details and fairness about the 2D baselines**

    - For Gen6D, OS2D, $\mathrm{OLN}_\mathrm{Corr.}$, and DTOID, we tried our best to re-train them using our OWID dataset for fairness. We have also tried directly using their official model trained on their official 2D dataset, while it performs worse than re-training with our new OWID dataset, e.g., see the Gen6D results on LM-O dataset as follows:

      | Training set\Metric |  mAR   |   AR50   |  AR75   |
    	| :-----------------: | :----: | :------: | :-----: |
      |        COCO         |  7.5   |   20.8   |   3.4   |
      |      **OWID**       | **12** | **26.8** | **4.8** |

    - For $\mathrm{OLN}_{\mathrm{DINO, CLIP}}$  we believe their pre-trained 2D representation from the large real-world dataset is good enough. So that we only used synthetic OWID to pre-train the open-world detection module and utilized the off-the-shelf CLIP and DINO models to match the reference images and proposals.

    - Following Reviewer deGE's valuable suggestion, ***we also included additional SOTA one/few-shot detectors, BHRL and Meta-DETR***, in our experiments. The comparison on the LM-O dataset is shown below, where VoxDet outperforms these class-level detectors by a large margin.

       | Model |  VoxDet  | BHRL | Meta-DETR |
       | :---: | :------: | :--: | :-------: |
       |  mAR  | **27.4** | 14.1 |   12.0    |
       | AR50  | **36.9** | 22.8 |   15.3    |
       | AR75  | **26.4** | 14.6 |   11.6    |

    To summarize, we have tried our best to ensure fairness in all experiments. We conclude that VoxDet outperforms these baselines by virtue of the innovative incorporation of explicit 3D geometry and reconstruction primitives. We will provide these details and experiments in our final version.

For the other specific concerns, we have tried to address them with detailed explanations, quantitative results, and qualitative visualizations (**in the Rebuttal PDF**) under each reviewer's comments area.

The authors are also open and ready for any further questions/suggestions in the discussion period.

---

> ### Comment · Area_Chair_v92Y · 2023-08-18
> **Post Rebuttal Feedback**
>
> Dear Authors,
>
> Thank you for this extensive and informative rebuttal. I think we have sufficient information regarding the reviewer feedback to come to a final decision among the reviewers and the AC.
>
> Best regards,
>
> Your AC

---

### Decision · Program_Chairs · 2023-09-21

**Decision:**

Accept (spotlight)

**Comment:**

The paper received positive ratings from all reviewers. It introduces an interesting 3D-aware instance detection method that is novel and demonstrates the promising properties of 3D-aware representations for detection, particularly in the context of novel instance detection